# Evaluation of the Rheologic and Physicochemical Properties of a Novel Hyaluronic Acid Filler Range with eXcellent Three-Dimensional Reticulation (XTR™) Technology

**DOI:** 10.3390/polym12081644

**Published:** 2020-07-24

**Authors:** Giovanni Salti, Salvatore Piero Fundarò

**Affiliations:** 1Medlight Institute, Via Monteverdi 2, 50144 Florence, Italy; 2Multimed Poliambulatorio e Day Surgery, Via dei Fornaciai 29/d, 40129 Bologna, Italy; dr.fundaro@gmail.com

**Keywords:** hyaluronic acid, physicochemical properties, rheology, XTR technology

## Abstract

Soft-tissue fillers made of hyaluronic acid and combined with lidocaine have recently become a popular tool in aesthetic medicine. Several manufacturers have developed their own proprietary formulae with varying manufacturing tools, concentrations, crosslinked three-dimensional network structures, pore size distributions of the fibrous networks, as well as cohesivity levels and rheological properties, lending fillers and filler ranges their unique properties and degradability profiles. One such range of hyaluronic acid fillers manufactured using the novel eXcellent three-dimensional reticulation (XTR™) technology was evaluated in comparison with other HA fillers and filler ranges by an independent research laboratory. Fillers manufactured with the XTR™ technology were shown to have characteristic rheological, crosslinking and biophysical factors that support the suitability of this filler range for certain patient profiles.

## 1. Introduction

Hyaluronic acid (HA)-based hydrogels have become the most commonly used soft-tissue fillers for facial rejuvenation [1]. Of all available filler material, only HA has fulfilled the need for a biologic substance that is easy to handle, safe, convenient and effective in addressing medically indicated conditions and has lower risk of inducing allergic response [2]. Manufacturers have also supplemented these characteristics by introducing novel HA filler formulations with the aim of stabilizing, increasing tissue longevity and enhancing their tolerability [3]. Distinct proprietary formulae have been developed using various manufacturing tools, HA concentrations, crosslinked three-dimensional network structures, pore size distributions of the fibrous HA networks, cohesivity levels and rheological properties. These properties influence the physical properties of the HA filler product and their variants, often referred to as ‘ranges’, which in turn affect their clinical effects. The most recognized of these formulae are the non-animal stabilized HA (NASHA^®^, (Galderma Pharma S.A., Lausanne, Switzerland)), Vycross^®^ (Allergan Inc., Irvine, CA, USA), cohesive polydensified matrix (CPM^®^, (Merz Pharmaceuticals GmbH, Frankfurt am Main, Germany)) and resilient HA (RHA^®^, (Teoxane Laboratories, Geneva, Switzerland)), among others [4,5,6].

Inherently, as the market for these products continues to grow, so does the need for a better understanding of the basic science of HA fillers [6]. No individual product can be used for every indication; hence, the clinician’s experience, as well as the specific guiding principles provided by the originating company, are often used as the bases for choosing the type of HA fillers that are appropriate for certain indications and for injection into specific anatomical layers. More importantly, whenever novel fillers and related technologies are introduced, developing a familiarity of the basic science of HA fillers would allow aesthetic medicine specialists to select the best options for their patients [7].

The main HA filler properties evaluated in literature include rheology, the degree of crosslinking, resistance to stretch and cohesivity [3,5,8] Together with proper selection of patients with ideal anatomy (e.g., appropriate skin thickness and with good anatomic architecture), knowledge of rheological properties can facilitate the selection of the right products that can help them achieve optimal aesthetic outcomes [5]. For instance, the degree and process of crosslinking changes the three-dimensional construction of the HA chains and affects the consistency of the filler. This altered structure may thus affect immune acceptance of the filler [5]. Cohesivity is an essential characteristic of gel implants and is defined as the capacity of a material not to dissociate. This property is naturally important during the product distribution into the tissues of the treated area and is considered necessary for the integrity of the solid and fluid phases of a gel. This is known to affect the lifting capacity of the HA fillers [3]. This allows for evaluation of the resistance to spreading and adhesion properties of the HA fillers. It also becomes possible to view the product’s homogeneity, which is often considered to equate to good dispersion qualities [3,5].

HA filler degradability by hyaluronidase can be considered as a “safety-feature” and gives a filler a potential edge over its competitors [9]. Technical errors, such as overfilling/overcorrection or filler placement at the wrong soft tissue plane or misplacement, may be reversed by hyaluronidase administration. Nonetheless, appropriate filler longevity in soft tissue is considered to provide structural stability and resiliency, which is a requirement for volume correction [10]. Thus, it is important for practitioners to find that delicate balance between stability and degradability to determine which fillers are best suited for their patients. Degradability studies have highlighted that HA content, cohesivity and crosslinking properties may play a role in the sensitivity of these fillers to enzymatic degradation [9,11].

In addition, one of the more important but often overlooked determinants of success in skin filler injections is control of pain. Preventive pain control helps not only minimize discomfort but also reduces procedural downtime while improving patient satisfaction. For this reason, the local anesthetic lidocaine has been incorporated in HA fillers. This widely accepted change has led to significant reduction in pain during and after injections. Furthermore, as a result of its antihistamine property, lidocaine reduces erythema, bruising and swelling [2]. However, data suggest that the presence of the local anesthetic drug can reduce the viscosity of the filler products. Hence, there is a need to evaluate the physical properties of HA fillers with adjunctive lidocaine [12].

In this publication, a recently developed line of soft-tissue fillers manufactured using the novel eXcellent three-dimensional reticulation (XTR™) technology have been subjected to the abovementioned tests in comparison with other HA fillers and filler ranges.

## 2. Methods

### 2.1. The XTR™ Technology Manufacturing Process

This novel HA filler range manufactured using the proprietary XTR™ technology (Definisse™, RELIFE S.r.l., Florence, Italy) was recently introduced globally. This technique employs a manufacturing process made up of a controlled three-step process broken down into the pretreatment, crosslinking and purification phases (Figure 1) [3]. The pretreatment phase is an initial fracturing step using a controlled thermal process. This leads to the bioengineered formation of size-specific HA fragments that have viscoelastic properties compatible to skin [3,13,14,15]. The process enables controlled cleavage of the length of the HA chains and is capable of obtaining two molecular weight ranges (medium: 2.5 × 10^6^–3.2 × 10^6^ Dalton, and high: 3.2 × 10^6^–3.5 × 10^6^ Dalton) [3]. A mixture of different lengths of HA chains are then intermolecularly bound by a crosslinking agent to produce a stable three-dimensional HA matrix [16]. The most commonly used chemical agent in creating inter- and intra-chain bridges between the HA biopolymer is 1,4-butanediol diglycidyl ether (BDDE) [16]. Crosslinking enables the HA molecules to have stable viscoelastic properties and allows for a low extrusion force when injecting the filler. It also improves filling properties and enhances the durability of the injectate [6]. Optimization of the cross-linking process enables minimal to almost zero free HA chains [3]. Lastly, the elimination of residual amounts of free unreacted BDDE and its byproducts is done via a series of downstream purification processes (i.e., precipitation, rehydration). Ultimately, the goal is to achieve minute amounts, i.e., <2 ppm, of remaining unreacted BDDE, a limit of detection recommended by the US Food and Drug Administration (FDA) [17]. BDDE levels are used as a parameter for testing and releasing each manufactured batch of fillers with XTR™ technology [3,18]. The size, crosslinking and purity of HA polymers used for filler materials is critical as these properties determine the biological effects of HA on skin tissue [13].

The biophysical properties of fillers with XTR™ technology in comparison with the most commonly available commercial fillers have been reviewed in this paper. It is expected that this study will provide an evidence-based scientific rationale for future investigations and clinical applications. The parameters tested in this study include rheology, crosslinking degree and cohesivity. These tests were conducted by the independent R&D Laboratory and Consultancy Center, Rigano Laboratories S.r.l. in Milan, Italy, from the period of June 2019 to July 2020.

The skin fillers tested included derivative HA ranges crosslinked with 1,4-butanediol diglycidyl ether (BDDE) and incorporated with 0.3% lidocaine (Table 1) [19,20].

### 2.2. Rheological Evaluation

Rheology tests were performed using a Rheoplus Anton Paar MCR 101 (Anton Paar Company, Graz, Austria) controlled rate rheometer equipped with a PP50-P2 sensor (50 mm parallel plates with serrated surfaces, gap 0.8 mm) and with a Peltier system for temperature control (i.e., 30 ± 0.05 °C). Before running the tests, the gels were allowed to rest for 300 s in order to provide a uniform temperature distribution and a common and consistent shear history amongst the samples [22].

The oscillatory shear stress test or the amplitude sweep test was done to measure the fillers’ linear elastic or storage modulus (G′) and the loss or viscous modulus (G′′), which are reflective of the elastic behavior and the flow of material when deformed, respectively. This was accomplished by varying the input shear strain from 0.01% to 4000% while keeping the frequency constant at 1 Hz on each of the samples. After the limit of the linear viscoelastic region (LVER) was identified, a frequency sweep test was performed. This was done to determine the mechanical spectra of the gels while varying the input frequency from 10 Hz to 0.01 Hz and keeping the shear strain constant within the LVER limit.

G′ measures the energy stored by the HA filler during deformation that re-establishes its original shape when the shear stress is removed. It corresponds to the elastic segment of its viscoelastic properties (or its semi-solid state). On the other hand, G′′ measures the energy lost on shear deformation because of internal friction. It characterizes the viscous segment of its viscoelastic properties (or the liquid state) of the sample. This property indicates the inability of the HA filler to completely recover its shape after the deformation. These two parameters are used to calculate the complex modulus (G*): √ (G′^2^ + G′′^2^). G* represents the total energy required to deform a gel using shear stress [23].

The loss factor (tan δ) is defined as the fraction between the loss modulus G′′ and the storage modulus G′. It indicates whether the material has mainly an elastic behavior or a viscous behavior, i.e., a tan δ < 1 means that the elastic component is more prominent in the gel structure [22]. In crosslinked HA fillers, tan δ usually ranges from 0.05 to 0.80; thus, elastic behavior under low shear stress is dominant over viscous behavior [23].

### 2.3. Degree of Crosslinking

The most common chemical HA crosslinking method uses ether formation by reaction of HA with 1,4-butanediol diglycidyl ether (BDDE) [24]. Under alkaline conditions, epoxides of BDDE react with the HA hydroxyl groups to form derivatives of 1,4-dibutanediol dipropan-2,3-diolyl ether (BDPE). Some of the BDPE form “true cross-links”, binding to HA at both ends, while others only bind HA at one end (“mono-linked”). To describe these HA filler modifications, Kenne and Wende and colleagues proposed terms that can be applied to characterize HA hydrogels crosslinked with BDDE [25,26]:The degree of modification (MoD) is the stoichiometric ratio between the sum of mono- and double-linked BDPE residues and HA disaccharide units. The more crosslink modifications seen when compared with the acetyl group, the higher the MoD%.The crosslinker ratio (CrR) indicates the fraction of double-linked crosslinker residues compared to all linked crosslinkers and this represents the measure of crosslinker efficiency.The degree of substitution (DS) is the proportion of the HA disaccharides that are substituted.The degree of crosslinking (CrD) is the stoichiometric ratio between BDPE residues that are double-linked and HA disaccharide units.The degree of crosslinking (DC) is the number of HA disaccharides involved in crosslinking in relation to the total number of HA disaccharides.

Kenne states that MoD and CrD quantify the total amount of linked BDPE and double-linked BDPE, respectively, in comparison to the total amount of HA. DS and DC are equivalent to the amount of the substituted HA and crosslinked HA, respectively, in comparison to the total amount of HA [26]. The relevance of MoD and DS is in describing the total change in the gel after modification. For example, a polymer with low MoD or DS is structurally similar with the original, intact gel. CrD and DC are more important in terms of describing the physical properties of the gel. A higher value of CrD for a gel reveals that it is a stronger gel (i.e., more crosslinked) and would swell less than a weaker gel with a lower CrD.

In this study, the determination of the degree of crosslinking of the nominated HA-based fillers was performed using the nuclear magnetic resonance (NMR)-based approach described by Wende and colleagues [25]. HA is hydrolyzed using an acidic medium, and then, the degradation products are analyzed using one-dimensional NMR. First, the samples were diluted to 4 mg/mL with acid water (HCl 0.1 M, pH 1.5), heated and gently mixed at 75 °C overnight. Once cooled, these samples were buffered at pH 7.0 with NaOH 1 M and 0.1 M, frozen and then lyophilized. After, the samples were dissolved in 0.5 mL of deuterated water (D_2_O) and then subjected to NMR Bruker 400 mHz analysis for the determination of MoD%, CrR and CrD. With the ^1^H-NMR set up, 64 scans and a recycle delay (D1) of 10 s were used. For the ^13^C-NMR set up, 8192 scans and a D1 of 10 s were used.

The three main parameters were then measured using the following formulae: (1) MoD (%) = (I ^δH1.5^/4)/(I ^δH1.9^/3) × 100, (2) CrR = 1-I^δC62.7^/(^Iδ25.2^/2) and (3) CrD (%) = (CrR × MoD) × 100.

### 2.4. Cohesivity

This study used the five-point visual Gavard-Sundaram Cohesivity Scale as a reference [27]. HA filler gel behaviors range from fully dispersed (non-cohesive) with only powder-like gel fragments visible to fully cohesive with only intact gel strands visible. Video documentation using standardized digital techniques was done to show how gels were extruded as a single filament from a syringe. Cohesivity of each specimen was visually assessed from these images as the ratio of intact gel to dispersed gel at each time point [27,28].

Cohesivity of HA filler calculated using the Gavard-Sundaram Cohesivity Scale indicates the affinity between the gel molecules and, according to the authors of the method, describes the tendency to maintain a homogeneous distribution within the dermis after injection [27]. It is used also to assess the ability of fillers to resist vertical compression/stretching and is the expression of the strength of internal forces that can be modified based on the HA concentration, the crosslinking technology and the different gel macrostructures (biphasic or monophasic gel) [4,23].

### 2.5. Degradability

Select samples were diluted in physiological buffer initially, and then, baseline readings of absorbance and relative wavelengths were measured using standard spectrophotometric analysis. Calibration curves (absorbance vs concentration) were calculated from the spectra from visible light to ultraviolet wavelength for each compound. A 1:8 dilution of all samples was designated as the appropriate concentration as described by La Gatta and colleagues [29].

The fillers were then incubated in the presence of bovine testicular hyaluronidase (BTH, Sigma Aldrich cat N. H3884) at 37 °C. The degradation was monitored by documenting the increase in soluble fraction amount during incubation. HA soluble fractions of each of the samples were then withdrawn and analyzed at different incubation times (i.e., 0, 30, 60, 300 min) as the ratio of HA concentration in the permeate versus the total HA concentration of the gel (%). Tests were performed at least in triplicate with the mean and standard deviation provided [30].

## 3. Results

The following results reflect the analyses of the physicochemical and mechanical properties of the selected injectable crosslinked HA gels based on previously developed pre-specified protocols.

### 3.1. Rheological Evaluation

The results of the viscoelastic measurements are shown in Figure 2 and Table 2 [22].

The rheological tests performed on the HA derivative gels highlighted considerable differences among the viscoelastic properties of each range and among fillers of the same group. Compared with the rest, NASHA appeared to have the highest elastic and viscous responses but also presented with the lowest critical strain (67%), which translates to less resistance to deformation. Each range is shown to gradually decrease in G′ down the line; lower G′ fillers are softer variants that are suited for medium to superficial implantation to the soft tissue. CPM-1 and CPM-2 presented a large LVER and the highest critical strain measurements (1200%), which indicates good strain resistance. The fillers, i.e., CPM, with higher tan δ are described to be more fluid than they are elastic. The ranges can be seen to have higher fluidity with lower HA concentrations. CPM-1 was the most viscous sample (tan δ = 0.49). The series of XTR, RL and CPM are composed of three variants each, with every sample having different properties, while VYC variants were observed to bear more similar intraclass rheological properties. XTR-2 and XTR-3 fillers had relatively higher G′ than their counterparts (Table 2). This may serve as an advantage in volumizing and lifting tissue and modulating muscular activity when injecting beneath the superficial musculoaponeurotic system (SMAS) layer but without the granular features of fillers with extremely high G′. Fillers with XTR™ technology in general have medium (intermediate) G′′ compared to the benchmarked products. [3].

### 3.2. Degree of Crosslinking

The crosslinking parameters are summarized in Table 3 [12].

Among the fillers, XTR-2 and XTR-3 and CPM-2 and CPM-3 presented with the highest MoD% values (i.e., 9% to 15%), which translates to being those with the highest degree of ether modification. MoD% for RHA-1 is about 3% and consequently is the filler with the lowest degree of modification. Other fillers presented intermediate values (i.e., 6–7.5%). CrR varied from 0.5 to 0.03. XTR-3, CPM-1, VYC-1 and VYC-3 had the highest CrR values, which denotes higher concentration of true crosslinkers compared to mono-linkers. CPM-3 and RHA-4 had the lowest CrR indicating that mono-linker linkages are more prevalent. The rest of the other fillers presented intermediate values.

### 3.3. Cohesivity

The HA samples provided were evaluated in terms of water-binding capacity via the validated Gavard-Sundaram Cohesivity Scale, where 1 means fully dispersed and 5 means fully cohesive at different time frames (Figure 3).

CPM gels had high cohesivity, being partially dispersed only at 70 s. CPM-1 was most dispersed at 70 s and maintained a very high cohesivity even after 1 min [28]. The other gels had an intermediate cohesivity and were mostly dispersed in as early as 30 s (i.e., VYC gels) or after 70 s (i.e., XTR). RL gels displayed high cohesivity at the first min and were dispersed only after a prolonged period. Gel dispersion among variants is shown in Figure 4.

### 3.4. Degradability

The mean soluble fractions (expressed in %) and the standard deviations observed through time were documented for the chosen fillers as follows (Table 4) [30]: 

Of the fillers evaluated, NASHA was observed to have the most rapid degradation, ultimately displaying a 95% soluble fraction after 30 min. XTR-1 had the lowest soluble fraction percentage at baseline, and this was seen to persist across the time points. XTR-2 and -3 were slightly more resistant to degradation over time, as shown in Figure 5.

## 4. Discussion

In this study, the aim was to determine the differences among rheological and physicochemical properties of fillers manufactured using XTR™ technology in comparison with other filler ranges of varying proprietary crosslinking technologies. Most of the fillers selected in this study have been used extensively in the past for various indications. In this simple series of laboratory evaluations, fillers manufactured using XTR™ technology were found to have distinct properties that set them apart from these filler ranges. The clinical implications of these unique characteristics are discussed.

### 4.1. Rheologic Properties

Exclusive manufacturing processes of HA fillers are used to alter HA molecular structure as well as their physicochemical and mechanical behaviors. These varied behaviors lend the product their unique characteristics and are perceived to affect their overall product performance [31]. Crosslinked HAs are viscous, so in order to facilitate extrusion of this material through a thin gauged needle, manufacturers have purposely produced low viscosity HA fillers with high shear rates. When extruded through a 30 G or 27 G needle, the fillers then become easier to inject [6].

G′, G′′ and G* describe the filler’s viscoelastic characteristics. Specific dynamic testing can be used to evaluate these rheological properties and resulting data not only indicate the physical/rheological features, but also provide rationale for the indications of the HA filler. Immediately after its injection and during its dwell into the soft tissues of the face, the HA filler is subjected both to shear stress, vertical compression and stretching forces, which lead to filler deformation. These forces also influence the fillers’ ability to correct volume loss and long-term tissue integration. G′ indicates the ability of HA fillers to rebound to their previous configuration after being altered by dynamic forces. The viscous modulus G′′ is a key parameter that characterizes HA resistance to these dynamic forces. G* measures the overall resistance to deformation of a gel, which comprises a recoverable component from the gel’s elasticity (G′) and a non-recoverable component from the gel’s viscosity (G′′) [32]. In general, the higher the G′, the higher is the elasticity and resistance to deformation and ultimately its capability to restore the soft tissue volume [31,33,34]. G′ also indicates filler stiffness or hardness and determines the injection at the appropriate anatomical plane. HA fillers with higher G′ have to be injected into the deep fat compartments or the pre-periosteal plane. These fillers have good volume restoration and can be injected deep into the soft tissue where they are neither palpable nor visible. Filler ranges have typically fulfilled this indication for volume correction of deep planes [31]. A potential advantage of high G′ is the capability to obtain a good prominent tissue projection and, with minimal amount of product, can provide structure to support the parts of the soft tissue with volume loss [3,35].

In this study, we evaluated both a biphasic filler (NASHA) and monophasic fillers (CPM, RL, VYC and XTR). To distinguish between these two filler types, we describe biphasic fillers as those with a solid phase (particles) admixed with a fluid phase such that when their physical characteristics, such as the particle size, are modified, it is possible to maintain a stable HA concentration. The monophasic fillers were originally thought to have more homogeneous structure and could hence be modified by changing the HA concentration and the degree of crosslinking [36]. Previous studies have shown that biphasic fillers usually have a higher G′ compared to monophasic fillers [37,38]. Note, however, that despite the term ‘monophasic’, these products still have observable gel particles and extractable HA much like ‘biphasic’ fillers [39].

In this study, we noted that, among the tested products, the biphasic NASHA filler had the highest G′ (571.61 Pa at 0.7 Hz), and among the monophasic fillers, the filler with a G′ value closer to NASHA was the XTR-3 (476.26 Pa at 0.7 Hz). These specific data indicate that among the tested monophasic fillers the XTR-3 has the highest elasticity and resistance to deformation. The XTR-3 filler is classified as a volumizer and compared with the other volumizer monophasic filler (VYC-3, RL-3, CPM-3) has the highest G′ (Figure 2A).

G′′ represents the energy lost on shear deformation due to internal friction. Sundaram described G′′ as a gel’s ability to dissipate energy when shear force is applied to it, such that it is equivalent to complex viscosity (η*)—a more useful parameter to describe viscous behavior—when shear force falls within the linear viscoelastic range [40]. G′′ in itself is not directly related to filler viscosity because HA filler is not purely viscous [23]. For these reasons, G′′ is not frequently used to describe the rheological characteristic of HA fillers. Just the same, a higher G′′ connotes greater thickness, requiring greater force for extrusion, while lower G′′ gels are easier to extrude through a fine needle.

Tan δ indicates if the filler has more gel-like or more water-like properties. The tan δ of HA fillers is always less than 1 and this indicates a predominance of an elastic behavior. A gel with high tan δ has a greater component of fluidity and a lower component of elasticity than a gel with low tan δ [23]. It is an indicator of elasticity and of capability to volume restoration of soft tissue and its value is inversely proportional to gel elasticity. Among the volumizer monophasic filler the lowest tan δ are observed in XTR-3 (0.063 at 0.7 Hz) and VYC-3 (0.067 at 0.7 Hz). These data confirm the observation previously described by G′, that is, the XTR-3 filler has a higher elasticity compared with the other monophasic volumizer fillers. The other two volumizer fillers (RL-3 and CMP-3) have a higher tan δ, which shows a good correlation with their G′ values. The tan δ difference between XTR vs VYC fillers and RL vs CMP fillers are also seen in their equivalent counterparts. In addition, a good correlation between tan δ and G′ can be gleaned from this observation (Figure 2).

Compared with the NASHA filler, all the newer filler ranges had lower G′, which provides these fillers a softer consistency and the versatility to be used for more indications. The fillers XTR-2 and XTR-3 had somewhat relatively higher G′ levels as their counterpart products from the other lines, namely, CPM-2, VYC-2 and RHA-3 for XTR-2, and CPM-3, RHA-4 and VYC-3 for XTR-3 (Table 2). The XTR fillers were also noted to have intermediate G′′, with XTR-2 having the highest G′′ value amongst them all. Having average G′′ with relatively higher G′ allows more prominent tissue projection as well optimal corrective effects with minimal amount of product [3]. The balance of G′ with other properties including cohesivity may have relevance in the fillers’ three-dimensional dynamic support [41].

XTR-1 has the highest tan δ, followed by XTR-2 and XTR-3. This means that XTR-1 has the highest capacity to flow when injected into the tissue while XTR-3 tends to resist flow. Relative to the corresponding fillers, XTR-2 and XTR-3 have greater tendency to be more elastic than fluid, which translates to good tissue support and volume efficiency for injecting close to the bone.

### 4.2. Crosslinking Properties

The degree of crosslinking denotes the mechanical strength of the gel, and hence, it represents a measure of product longevity. BDDE is essential in optimizing the crosslinking of HA. However, residual BDDE may have deleterious effects on the human body as a result of the reactive epoxide groups found in the free BDDE molecules; hence, the standard industry specification of <2 ppm of BDDE in the HA product has become a requirement by US FDA standards [17]. Reportedly, products with an excessive degree of crosslinking have been found to have increased reactivity and may also increase the risk of inflammation and granuloma formation [42]. Nonetheless, a balanced degree of crosslinking may be beneficial in reducing the material swelling of the filler [43].

XTR-2 and XTR-3 are among the fillers with the highest degree of crosslinking compared with the other nominated filler products. Additionally, XTR-3 has the highest degree of crosslinking of 5%, which denotes optimal modification and effective crosslinking. Clinically, these properties are perceived to provide the product the highest level of longevity for application in deeper planes of the face.

### 4.3. Cohesivity

Unlike rheology, there are currently no ready-made instruments designed or methods which could easily be used to measure this variable. In effect, the scientific opinions concerning its clinical relevance are conflicting [36,44]. Furthermore, other than the Gavard-Sundaram Cohesivity Scale [27], a few other non-standardized tools, such as the linear compression test [4] and the average drop-weight method, are also used [44]. These tests are not known to provide consistent data; however, all cohesivity measurement methods indicate the affinity between the gel molecules determined by the internal adhesion forces due to the crosslinking agent. Fillers implanted into deep anatomic layers of the face are constantly subjected to compression forces and tension from outside forces. Fillers with high cohesivity can resist vertical compression and have a greater capability to maintain their original shape after injection. These fillers can maintain a homogeneous distribution within the dermis. Fillers with intermediate cohesivity distribute with a transitional pattern and fillers with low cohesivity have a higher tendency to have intradermal microbolus dispersion. Fagien et al. presented data that suggest that as G′ decreases, the gel may exhibit more cohesive properties [31]. The same inverse relationship was noted by Edsman et al. [44] further supporting the observation that this relationship appeared to exist only among products made by the same technology [31]. Hence, when comparing the lasting effects of two HA fillers with the same G′ but different cohesivity, the filler with lower cohesivity loses tissue projection more easily than fillers with higher cohesivity.

Our study likewise presents this inverse correlation between G′ and cohesivity: fillers with higher G′ (NASHA) have lower cohesivity, fillers with intermediate G′ (XTR and VYC) have intermediate cohesivity and fillers with lower G′ (RL and CPM) have higher cohesivity. The NASHA filler having a lower cohesivity may be attributed to its biphasic structural characteristic.

G′ and cohesivity are related to the capability of soft tissue volume restoration of the fillers, and there appears to be a relationship between them where one compensates for the lack of the other. The two characteristics determine the behavior of the fillers after injection and therefore also the clinical results. The mechanistic relationship between these has yet to be established by further studies.

Our study partially confirms the inverse relationship between cohesivity and G′. Cohesivity within the groups of XTR and VYC fillers is stable. On the other hand, in the RL and CPM groups, cohesivity increases with lower G′ (Figure 4D). The cohesivity data of NASHA, VYC and CPM fillers confirm the behavior described in the pilot validation study of the Gavard-Sundaram Cohesivity Scale [27].

Among the monophasic fillers, the XTR group had the higher G′ and an intermediate cohesivity, while the VYC group has lower G′ and lower cohesivity, and the RL and CPM groups have significantly lower G′ and higher cohesivity. This data might indicate a good volumizing capability by XTR fillers compared with other fillers evaluated in this study, but further research is necessary to establish this.

### 4.4. Degradability

The general assumption is that fillers with higher HA concentration and stronger degree of crosslinking are more likely to be resistant to enzymatic degradation. Therefore, additional hyaluronidase sessions are usually needed for patients receiving fillers with higher HA concentration [10]. However, different studies have demonstrated contradictory results attributed to disparities in hyaluronidase source (bovine, ovine, recombinant human), doses used, as well as diverse experimental formats [9].

Results of this stability study are consistent with the crosslinking evaluation, such that XTR-2 and XTR-3 with the highest degree of crosslinking were shown to be more stable to degradation than NASHA after 5 h of hyaluronidase exposure. Further exploration to a prolonged exposure time, evaluation of a dose-dependent response and use of a broader range of hyaluronidases in vivo may provide more insights to these observations.

### 4.5. Clinical Application, Limitations and Recommendations

Filler ranges come in a series of increasing concentrations, which are supposed to distinguish their application and purpose. Interestingly, concentration alone does not by itself control filler performance. Filler rheological properties, namely the elastic modulus, viscous modulus, tan δ and cohesivity, along with crosslinking properties, manufacturing processes and injector technique are observed to more likely predict the longevity and persistence of HA fillers in tissue [38]. All these features contribute to defining the global physical characteristics of the HA filler, whose sum of all properties determines its capability to modify soft tissue. In addition, ‘higher’ does not always necessarily translate to ‘better’—such that a filler with higher G′ may have greater strength but does not necessarily have a greater ability to lift [45]. Therefore, it is important to interpret these biophysical factors holistically, together with their tissue integration and biostimulatory capabilities in perfect balance to determine their suitability in patients with certain profiles (i.e., with specific tissue and anatomical characteristics).

When using the products with XTR™ technology, the general suggestion is to inject less than usual to avoid instilling excessive deposits in the soft tissue. The stronger gels, XTR-2 and XTR-3, are expected to produce long-term and prominent projection with the use of only small amounts of product in both subcutaneous and supraperiosteal injections. The sequence of G′ and cohesivity of the fillers made with XTR™ technology can be observed to be consistent with the indicated depths of injection, as well as their clinical application.

The current data is limited because of the relatively short period that XTR™ Technology has been in the market. More long-term efficacy and safety studies are expected to provide additional insights to the clinical advantages of the properties evaluated in this study. We also aim to conduct further studies on the biodegradability, in vivo kinetics, biostimulatory properties and tissue integration of the fillers to ensure a comprehensive evaluation. This premier study of physicochemical and rheological properties of fillers manufactured using XTR™ technology evaluated through laboratory testing is expected to exemplify an evidence-based approach in the selection of fillers. This, in turn, will help in the procedural application of fillers, ensuring optimal outcomes in volume restoration, improvement of volume distribution and, ultimately, achieving a balanced facial contour.

## 5. Conclusions

An evidence-based approach to selection of fillers based on rheologic and other physicochemical properties, combined with proper technique, may facilitate the choice of the best product, achieving optimal results. Since the incorporation of local anesthetic has become standard practice in filler manufacturing, there is a need for exploration of the physicochemical properties of these fillers upon integration in soft tissue. Lastly, owing to the findings reported in this article, fillers with XTR™ technology may be a viable option for patients wanting an option with lasting lifting and rejuvenating effects.

## Figures and Tables

**Figure 1 polymers-12-01644-f001:**
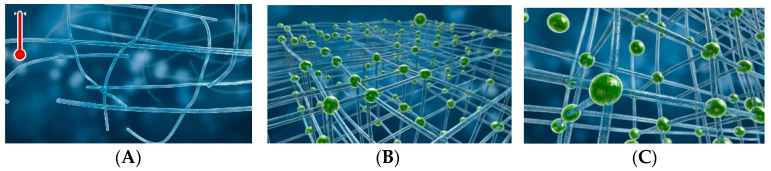
The eXcellent three-dimensional reticulation (XTR™) technology manufacturing process. (**A**) The pretreatment phase. (**B**) The crosslinking phase. (**C**) The purification process.

**Figure 2 polymers-12-01644-f002:**
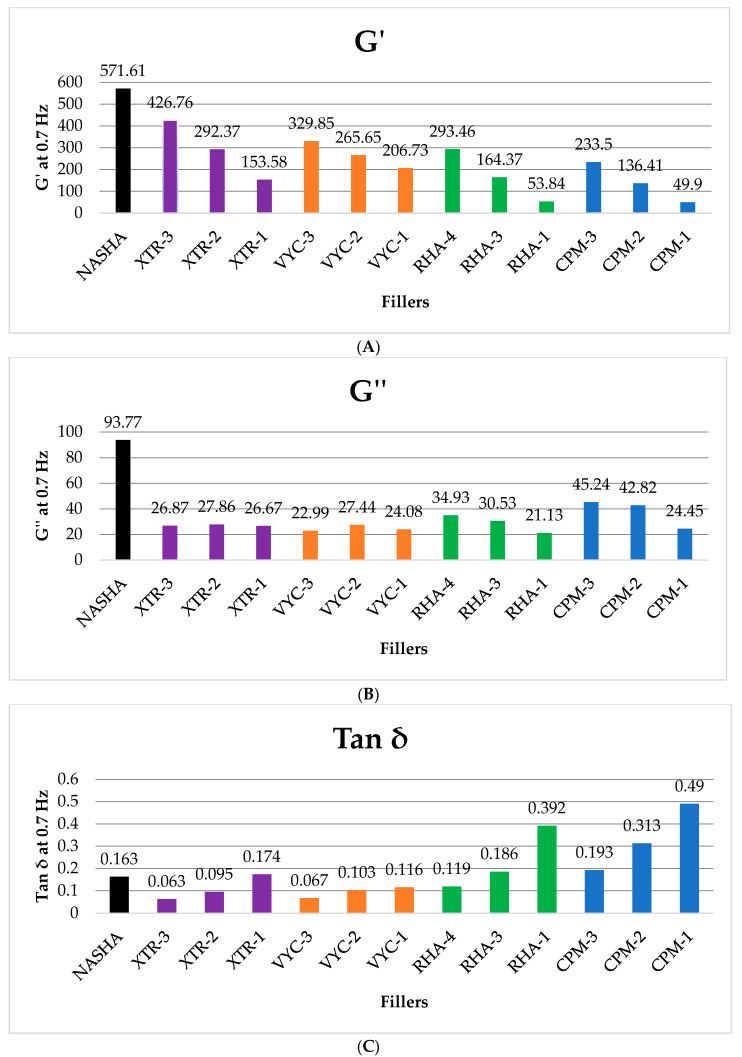
Summary of frequency sweep and amplitude sweep measurements of the HA fillers evaluated. A comparison of the **(A)** elastic moduli (G′), **(B)** viscous moduli (G′′), and **(C)** tan δ of fillers made with varying crosslinking technologies.

**Figure 3 polymers-12-01644-f003:**
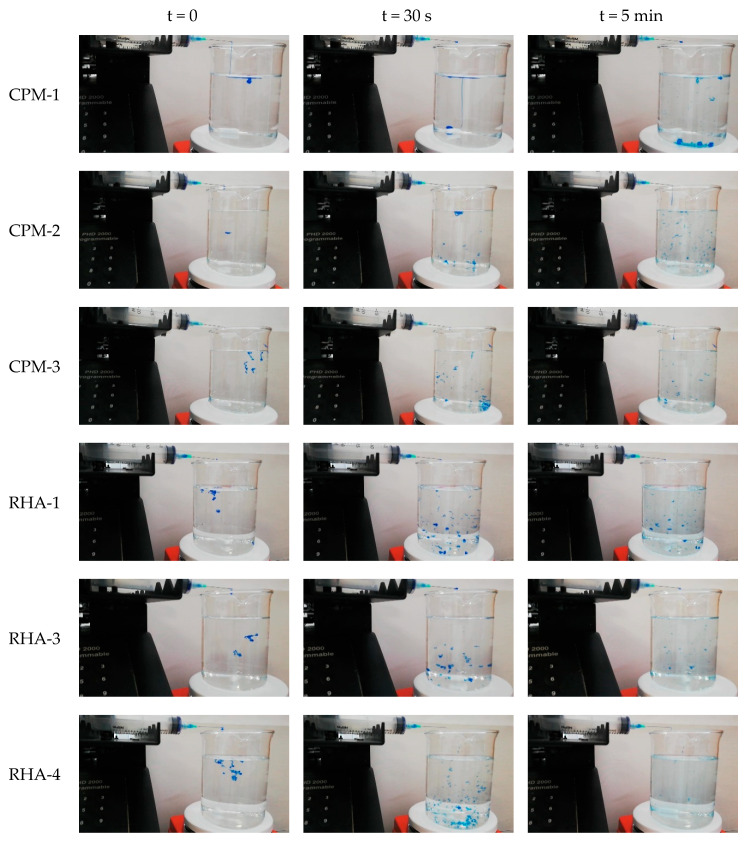
Gel dispersion in water at T = 0, 30 s, and 5 min of evaluated fillers.

**Figure 4 polymers-12-01644-f004:**
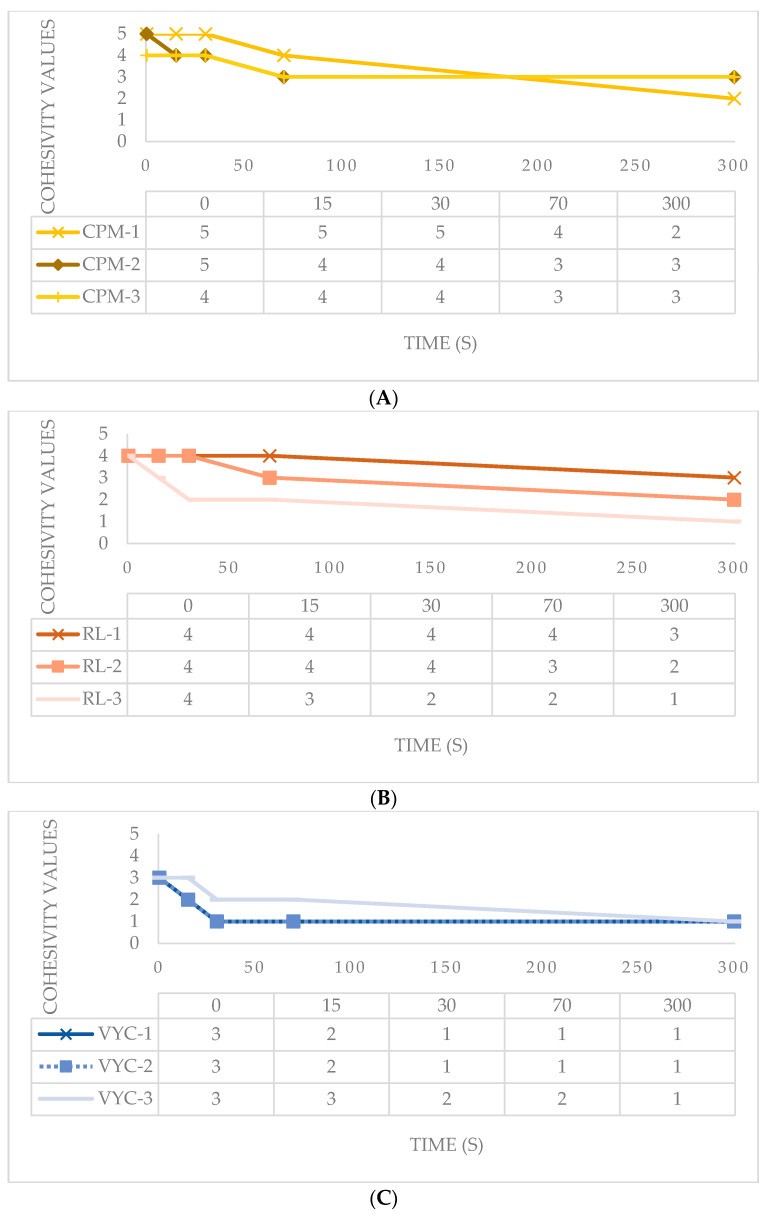
Gel dispersion in water at all time points for HA fillers evaluated. (**A**) CPM gels; (**B**) RL gels; (**C**) VYC gels; (**D**) XTR gels.

**Figure 5 polymers-12-01644-f005:**
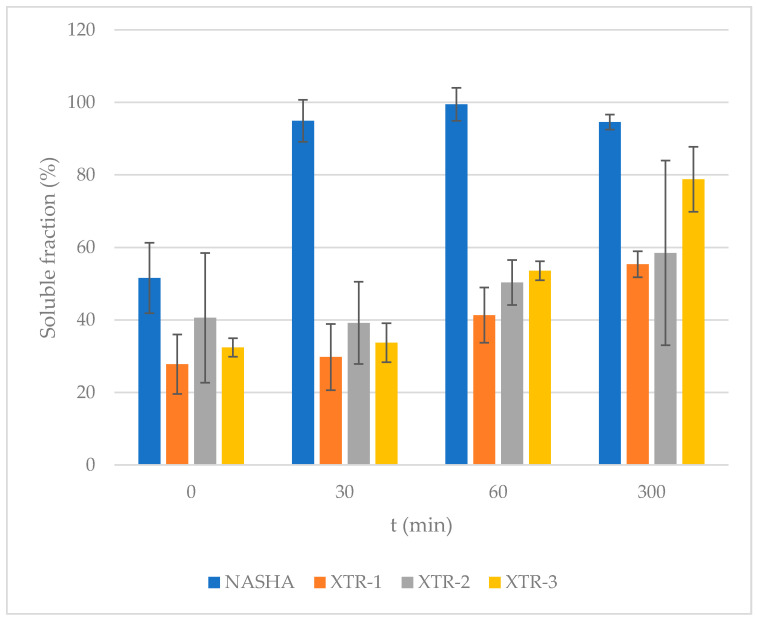
Percentage of soluble fraction for each gel type (mean and standard deviation).

**Table 1 polymers-12-01644-t001:** Hyaluronic acid (HA) gel fillers analyzed for biophysical properties.

Filler Identifier	Crosslinking Technology [20]	Hyaluronic Acid Concentration (mg/mL) [19,20,21]
NASHA	NASHA^®^	20
CPM-1	CPM^®^	22.5
CPM-2		25
CPM-3		26
RHA-1	RHA^®^	15
RHA-3		23
RHA-4		23
VYC-1	Vycross^®^	15
VYC-2		17.5
VYC-3		20
XTR-1	XTR™	23
XTR-2		23
XTR-3		25

**Table 2 polymers-12-01644-t002:** Comparison of representative fillers from each range with similar HA concentrations.

Product	G′	G′′	Tan δ
XTR-1	153.58	26.67	0.174
CPM-1	49.9	24.45	0.49
VYC-1	206.73	24.08	0.116
RHA-1	53.84	21.13	0.392
XTR-2	292.37	27.86	0.095
CPM-2	136.41	42.82	0.313
VYC-2	265.65	27.44	0.103
RHA-3	164.37	30.53	0.186
XTR-3	426.76	26.87	0.063
CPM-3	233.5	45.24	0.193
RHA-4	293.46	34.93	0.119
VYC-3	329.85	22.99	0.067

**Table 3 polymers-12-01644-t003:** Summary of crosslinking parameter measurements of the HA fillers evaluated.

Product	mg/mL HA	MoD%	CrR	CrD%
CPM-1	22.5	7.50	0.48	3.57
CPM-2	25.5	9.80	0.11	1.06
CPM-3	26	15.90	0.03	0.41
RHA-1	15	3.16	0.25	0.80
RHA-3	23	6.02	0.31	1.87
RHA-4	23	6.85	0.09	0.64
VYC-1	15	6.61	0.43	2.85
VYC-2	17.5	7.73	0.14	1.08
VYC-3	20	7.36	0.45	3.31
XTR-1	23	7.01	0.15	1.05
XTR-2	23	12.64	0.27	3.35
XTR-3	25	10.16	0.49	5.00

MoD, degree of modification; CrR, crosslinker ratio; CrD%, degree of crosslinking.

**Table 4 polymers-12-01644-t004:** Soluble fractions (mean and standard deviation) for each of the HA fillers evaluated.

		Enzymatic Degradation Time			
Product	mg/mL HA	t = 0	t = 30 min	t = 1 h	t = 5 h
NASHA	20	51.58 (9.71)	94.95 (8.19)	99.50 (17.89)	94.56 (2.54)
XTR-1	23	27.8 (5.79)	29.79 (9.14)	41.34 (11.35)	55.36 (5.39)
XTR-2	23	40.59 (4.55)	39.20 (7.62)	50.35 (6.23)	58.47 (2.61)
XTR-3	25	32.43 (2.08)	33.72 (3.59)	53.58 (25.46)	78.79 (8.95)

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
