# Peer review of "Evaluation of the Rheologic and Physicochemical Properties of a Novel Hyaluronic Acid Filler Range with eXcellent Three-Dimensional Reticulation (XTR™) Technology"

_polymers, 2020, doi:10.3390/polym12081644_

Round 1

Reviewer 1 Report

The paper entitled ‘Evaluation of the rheologic and physicochemical properties of a novel hyaluronic acid filler range with eXcellent Three-dimensional Reticulation (XTR™) technology’ covers a very interesting and applicable topic. The authors study the line of soft-tissue fillers with lidocaine to enhanced stability and obtain the longer-lasting cosmetic effects. Before the recommendation to publish this work, I would like to ask the authors for a few clarifications on the following points:

  1. Figure 1 missing figure caption. I am not sure what is the difference in the image with the description crosslinking phase and purification process. It is just the zoom in to the previous image?
  2. Figure 2 missing the description of a), b) and c). Add also the axis and tick on the presented graphs so it will be easier to follow the rheological changes in fillers
  3. Figure 3 – the dispersion? Any other test to validated or just based on these images?
  4. Some part needs to be rewritten as it is not clear what authors want to say…
  5. Use SI units s=seconds, min= minutes throughout the paper
  6. Figure 4 missing a) b)… and the description in the figure caption. What are the units on X and Y axis? Can you add axis and reformat the graphs so they are easier to follow the results? The yellow line in the two graphs is missing or is invisible?
  7. Line 224 not sure if that statement is necessary ‘mechanical properties of a novel range of fillers manufacture using XTR™ ‘
  8. Line 270: ‘Sundaram described G'' as gel’ – missing reference number after the name?
  9. Can you define the best injection properties? Of course, all change once it is done under the skin as the dispersion is very different from what you present in Fig. 3. Could add the discussion points considering these issues.
  10. Line 302: ‘ volume efficiency for use in deeper injections’ – how deep are eventually the injections to which part of skin?
  11. The separate ‘conclusions’ section is missing.

Author Response

We sincerely thank you for your invaluable feedback. Please see our point-by-point response to the queries you have sent us. Thank you and we are happy to provide further information, if needed. We believe that it is important for the community to initiate studies on hyaluronic acid, which has become the standard filler of choice by many clinicians. Although there are studies on the rheology and viscoelasticity of lidocaine combined with fillers, these are mostly on filler materials that are already less used at this time (i.e. calcium hydroxyapatite). Therefore, we have set forth to investigate physical properties of HA with lidocaine, especially the new crosslinked formulation made with XTR Technology.

The paper entitled ‘Evaluation of the rheologic and physicochemical properties of a novel hyaluronic acid filler range with eXcellent Three-dimensional Reticulation (XTR™) technology’ covers a very interesting and applicable topic. The authors study the line of soft-tissue fillers with lidocaine to enhanced stability and obtain the longer-lasting cosmetic effects. Before the recommendation to publish this work, I would like to ask the authors for a few clarifications on the following points:

Point 1: Figure 1 missing figure caption. I am not sure what is the difference in the image with the description crosslinking phase and purification process. It is just the zoom in to the previous image?

Thank you for letting us know. We have already placed the figure title “Figure 1. The XTR™ Technology manufacturing process”. In this figure, we had hoped to depict the purification process (3rd image) with an image zoomed in, where there should less of the free floating BDDE. Also, we wanted to illustrate that fillers manufactured with XTR Technology comply with the US FDA recommended limit of unreacted BDDE.

Point 2: Figure 2 missing the description of a), b) and c). Add also the axis and tick on the presented graphs so it will be easier to follow the rheological changes in fillers

Thank you. There are now text descriptions for A, B and C, as well as gridlines in the graphs to facilitate comparison of the bars. 

Point 3. Figure 3 – the dispersion? Any other test to validated or just based on these images?

Yes, we have performed the Gavard-Sundaram Cohesivity Scale, which is a validated scoring system used by many laboratories for measuring cohesivity of fillers. Although cohesivity is one of the main physicochemical properties of fillers measured in our study, it is not in and of itself our basis for comparison. There is still a need to correlate this with other tests, such as the rheological study, to determine the behaviour of the fillers.

Point 4. Some part needs to be rewritten as it is not clear what authors want to say…

This is noted. We have reworded the portions that needed some clarity.

Point 5. Use SI units s=seconds, min= minutes throughout the paper

The correction is well noted. All units have been amended to reflect this direction.

Point 6. Figure 4 missing a) b)… and the description in the figure caption. What are the units on X and Y axis? Can you add axis and reformat the graphs so they are easier to follow the results? The yellow line in the two graphs is missing or is invisible?

Thank you. The graphs have been amended to show the axes. The yellow lines in Figure 4 C and D were overlapping with the blue and green lines, respectively. We have fixed the lines to show dashed lines on top of solid lines.

Point 7: Line 224 not sure if that statement is necessary ‘mechanical properties of a novel range of fillers manufacture using XTR™ ‘

The new sentence reads: “In this study, the aim was to determine the differences among physicochemical properties of fillers manufactured using XTR™ Technology in comparison with other filler ranges of varying proprietary crosslinking technologies.”

Point 8: Line 270: ‘Sundaram described G'' as gel’ – missing reference number after the name?

We have included the reference for this. Thank you.

Point 9: Can you define the best injection properties? Of course, all change once it is done under the skin as the dispersion is very different from what you present in Fig. 3. Could add the discussion points considering these issues.

Clinically, physicochemical evaluation is only intended to describe the fillers and derive information on the specific indications for which they are to be used. The tests are not used to determine the ‘best’ filler properties and are instead conducted to show where they are suited for use (i.e. rigid support for deep tissues vs. biostimulation or correction of superficial fine lines and creases). We enhanced the discussion section to correlate the findings with dispersion and the potential applications of the products.

Point 10: Line 302: ‘ volume efficiency for use in deeper injections’ – how deep are eventually the injections to which part of skin?

We have specified the region and the layer of the facial tissue. “Relative to the corresponding fillers, XTR-2 and XTR-3 has the tendency to be more elastic than they are fluid, which translates to good tissue support and volume efficiency for use in injections of close to the bone at the area of the mid- and lower face.”

Point 11: The separate ‘conclusions’ section is missing.

Thank you. We have made a separate heading for the conclusions.

Reviewer 2 Report

MS ID: Polymers 2020, 12, x; doi: FOR PEER REVIEW

Title: Evaluation of the rheologic and physicochemical properties of a novel hyaluronic acid filler range with excellent three-dimensional reticulation (XTR™) technology.

Authors: Salti G.,, Fundarò S.

General Comments:

This hyaluronic acid soft-tissue fillers seem to have many beneficial properties and could be very useful in many clinical aspects. Even though it has been cross –linked, I still have much concern about its biodegradability. For this the authors is requested to make some comments.  

Specific Comment:

  1. Line 74-105/Section 2:

Please reformat the whole paragraph. Be advised that your description must aim at increasing the readability and readership. This submitted version as it is in this manuscript has not been clearly edited regarding the preparation procedure.

  1. Section 3.3: Cohexivity

     Please use lower ‘t’ to indicate ‘time’. . Upper “T” is usually used for denoting ‘temperature’.

  1. Line 95-97: What does this graphic illustration mean? I could not catch your meaning. On the other hand, I am very interested in the purification process. Please present it in detail the purification technology, the analytical data except the rheological analysis that you have shown in the following section.

Author Response

Reviewer 2:

We sincerely thank you for your invaluable feedback. Please see our point-by-point response to the queries you have sent us. Thank you and we are happy to provide further information, if needed. We believe that it is important for the community to initiate studies on hyaluronic acid, which has become the standard filler of choice by many clinicians. Although there are studies on the rheology and viscoelasticity of lidocaine combined with fillers, these are mostly on filler materials that are already less used at this time (i.e. calcium hydroxyapatite). Therefore, we have set forth to investigate physical properties of HA with lidocaine, especially the new crosslinked formulation made with XTR Technology.

MS ID: Polymers 2020, 12, x; doi: FOR PEER REVIEW

Title: Evaluation of the rheologic and physicochemical properties of a novel hyaluronic acid filler range with excellent three-dimensional reticulation (XTR™) technology.

Authors: Salti G.,, Fundarò S.

General Comments:

Point 1: This hyaluronic acid soft-tissue fillers seem to have many beneficial properties and could be very useful in many clinical aspects. Even though it has been cross –linked, I still have much concern about its biodegradability. For this the authors is requested to make some comments.

Thank you kindly for the feedback. This is the first publication on physicochemical properties of the fillers manufactured with XTR technology. Other properties of the fillers, such as its long-term stability, biodegradability, in vivo kinetics, biostimulatory effects, and tissue integration, which are all vital for characterisation of soft tissue fillers, may be evaluated in subsequent studies (Mochizuki M, et al. Plast Reconstr Surg. 2018 Jul;142(1):112-121.)

Specific Comment:

Line 74-105/Section 2:

Point 2: Please reformat the whole paragraph. Be advised that your description must aim at increasing the readability and readership. This submitted version as it is in this manuscript has not been clearly edited regarding the preparation procedure.

Thank you kindly for the feedback. The text was supposed to be labels of the processes described in figure 1; hence, we moved these texts to the bottom of the corresponding figures. We hope that this change was able to address the matter.

Section 3.3: Cohesivity

Point 3: Please use lower ‘t’ to indicate ‘time’. . Upper “T” is usually used for denoting ‘temperature’.

Thank you and duly noted. We have revised ‘T’ to ‘t’ to indicate time. We appreciate the feedback.

Point 4: Line 95-97: What does this graphic illustration mean? I could not catch your meaning. On the other hand, I am very interested in the purification process. Please present it in detail the purification technology, the analytical data except the rheological analysis that you have shown in the following section.

Thank you for the comments. We believe this is addressed by moving the figure labels and placing them where they were intended under each picture A, B and C. We have expounded on the purification process as shown in the figure description of figure 1C.

Reviewer 3 Report

In this manuscript, the authors described the rheological and physiochemical properties of a hyaluronic acid fillers manufactured with a novel reticulation technology named (XTRTM) in comparison with other HA fillers commercially available. The manuscript is not so clear and most of all it is difficult to understand if the authors is proprietary of the XTR technology or they just compare the DefinisseTM Fillers with the other present on the market. Moreover, the manuscript has low scientific soundness because it only compare the physicochemical properties of some commercial fillers without adding anything to the actual knowledge about hyaluronic acid fillers. The characterization is not complete and I would suggest performing some degradation analysis to understand how the mechanical and rheological properties of the fillers change in biological environment with time. The conclusion are so vague and with low scientific soundness because they add very little to the actual knowledge about HA fillers. Furthermore, it is difficult to understand the final goal of the manuscript. I would suggest the acceptance of the present manuscript in Polymers Journal after some major revisions that I report below:

  • Label of figure 1 is missing.
  • At pp 3 line 98, I would not use the term in vitro for the biophysical properties because it is usually referred to the biological evaluation of materials.
  • The authors compared a biphasic filler (NASHA) with monophasic fillers (CPM, RL, VYC and XTR) but in my opinion, the presence of a second phase will modify the rheological characteristic of the fillers making it not comparable with the monophasic fillers.
  • I would better explain the various method to define the degree of crosslinking. In particular the second, the DS, it is not explained how can be measured the degree of substitution of HA.
  • The cohesivity study is a qualitative test most based on visible behaviour with ambiguous physical meaning. The water binding capacity is a value with completely subjective description that could change from different operators. In my opinion, the test has no scientific relevance and I would suggest testing the dynamic mechanical behaviour of the filler in biological environment at different time.
  • Did the author evaluate the degradation behaviour of the proposed fillers?
  • The excellent XTRTM reticulation technologies does not provide any evidence of enhanced stability and clinical long lasting effect. In fact, the Gel dispersion test the XTR gels has lower values if compared with CPM gels and RL gels.
  • The statement at pp13 from line 373 to line 378 has no sense and sounds more like an advertisement of the product that as a scientific description. I would delete this part.

Author Response

Reviewer 3:

We sincerely thank you for your invaluable feedback. Please see our point-by-point response to the queries you have sent us. Thank you and we are happy to provide further information, if needed. We believe that it is important for the community to initiate studies on hyaluronic acid, which has become the standard filler of choice by many clinicians. Although there are studies on the rheology and viscoelasticity of lidocaine combined with fillers, these are mostly on filler materials that are already less used at this time (i.e. calcium hydroxyapatite). Therefore, we have set forth to investigate physical properties of HA with lidocaine, especially the new crosslinked formulation made with XTR Technology.

In this manuscript, the authors described the rheological and physiochemical properties of a hyaluronic acid fillers manufactured with a novel reticulation technology named (XTRTM) in comparison with other HA fillers commercially available. The manuscript is not so clear and most of all it is difficult to understand if the authors is proprietary of the XTR technology or they just compare the DefinisseTM Fillers with the other present on the market. Moreover, the manuscript has low scientific soundness because it only compare the physicochemical properties of some commercial fillers without adding anything to the actual knowledge about hyaluronic acid fillers. The characterization is not complete and I would suggest performing some degradation analysis to understand how the mechanical and rheological properties of the fillers change in biological environment with time. The conclusion are so vague and with low scientific soundness because they add very little to the actual knowledge about HA fillers. Furthermore, it is difficult to understand the final goal of the manuscript. I would suggest the acceptance of the present manuscript in Polymers Journal after some major revisions that I report below:

Point 1: Label of figure 1 is missing.

Thank you for the feedback. We have moved lines 74-105 to the bottom of the figure as these lines were intended to be figure labels.

Point 2: At pp 3 line 98, I would not use the term in vitro for the biophysical properties because it is usually referred to the biological evaluation of materials.

The comment is well received. We have removed the term in vitro in this line and in other parts.

Point 3: The authors compared a biphasic filler (NASHA) with monophasic fillers (CPM, RL, VYC and XTR) but in my opinion, the presence of a second phase will modify the rheological characteristic of the fillers making it not comparable with the monophasic fillers.

Although the term ‘monophasic’ is used to refer to these fillers, these products still have observable gel particles and extractable HA much like ‘biphasic’ fillers (Öhrlund JÅ, Edsman KL. The Myth of the "Biphasic" Hyaluronic Acid Filler. Dermatol Surg. 2015 Dec;41 Suppl 1:S358-64). The technology used for mono- and biphasic HA fillers may be different providing these fillers different rheological characteristics; however, both types are used for the same clinical purpose. In general, biphasic fillers have higher G’, lower tan δ and lower cohesivity; monophasic fillers have lower G’, higher tan δ and higher cohesivity. In this study, the comparison between monophasic fillers and biphasic fillers allowed us to show that fillers manufactured with XTR™ technology have G’ values that approximate those of the biphasic filler but have cohesive properties that are comparable with other monophasic fillers.

Point 4: I would better explain the various method to define the degree of crosslinking. In particular the second, the DS, it is not explained how can be measured the degree of substitution of HA.

Thank you for your comment. We have added the following statements regarding degree of substitution: “The relevance of MoD and DS is in describing the total change in the gel after modification. For example, a polymer with low MoD or DS is structurally similar with the original, intact gel. CrD and DC are more important in terms of describing the physical properties of the gel. A higher value of CrD for a gel reveals that it is a stronger gel (i.e. more crosslinked) and would swell less than a weaker gel with a lower CrD.”

Point 5: The cohesivity study is a qualitative test most based on visible behaviour with ambiguous physical meaning. The water binding capacity is a value with completely subjective description that could change from different operators. In my opinion, the test has no scientific relevance and I would suggest testing the dynamic mechanical behaviour of the filler in biological environment at different time.

Thank you for pointing these out. The Gavard-Sundaram Scale is a tool being used in many laboratories to evaluate cohesivity and has been validated in several studies. Although subjective and has no direct correlation with fillers’ clinical properties, it can still “provide a scientific rationale for the intuitive selection of different products for specific clinical objectives,” according to the authors of the scale. Additionally, cohesivity is not in and of itself our basis for our evaluation of filler properties. There is still a need to correlate it to other characteristics such as the rheology of the filler, its crosslinking, swelling properties, and others to determine the suitability of the filler for a specific indication.

Point 6: Did the author evaluate the degradation behaviour of the proposed fillers?

We have yet to conduct these studies and are planning to evaluate the filler biodegradability and its other behavior upon injection to the skin in an upcoming publication. This manuscript is currently a preliminary evaluation of the filler based on the evidence that is currently available. Additionally, we have mentioned these limitations within the discussion section: “The current data is limited because of the relatively short period that XTR™ Technology has been in the market. More long-term efficacy and safety studies are expected to provide additional insights to the clinical advantages of these properties. We also aim to conduct further studies on the biodegradability, in vivo kinetics, biostimulatory properties, and tissue integration of the fillers to ensure a comprehensive evaluation. This premier study of physicochemical and rheological properties fillers manufactured using XTR™ Technology evaluated through laboratory testing is expected to exemplify an evidence-based approach in the selection of fillers. This, in turn, will help in the procedural application of fillers ensuring optimal outcomes in volume restoration, improvement of volume distribution, and ultimately, a balanced facial contour.”

Point 7: The excellent XTRTM reticulation technologies does not provide any evidence of enhanced stability and clinical long lasting effect. In fact, the Gel dispersion test the XTR gels has lower values if compared with CPM gels and RL gels.

The gel dispersion test evaluates cohesivity. As mentioned earlier, there is a need for the evaluation of other properties to provide a bigger picture of how fillers behave. There is no direct correlation between the gel dispersion test and the filler’s stability or duration. The XTR reticulation technology modifies all the rheological characteristics and the interplay among these characteristics lends features that are unique to this filler: high G' and medium cohesivity. The CPM and RL gels have higher cohesivity but lower G'.

Point 9: The statement at pp13 from line 373 to line 378 has no sense and sounds more like an advertisement of the product that as a scientific description. I would delete this part.

Thank you and the feedback is well received. We have removed these lines in the resubmitted draft.

Round 2

Reviewer 2 Report

No further comments!

Author Response

Thank you kindly for your comments. We truly appreciate your constructive feedback.

Reviewer 3 Report

The authors replied to all the reviewer comments but they did not add some required experiments. In my opinion, they do not increase the scientific soundness of the manuscript.

Point 1: Ok.

Point 2: Ok.

Point 3: Ok.

Point 4: the Authors just added better description of the method to define the degree of crosslinking. However, I would ask a better explanation about the experimental procedure to determine the different degree of crosslinking. 

Point 5: Ok.

Point 6: I believe that the degradation studies are fundamental to define the properties of a HA a filler. Therefore, I continue to say that degradation test is fundamental to increase the scientific soundness of this manuscript.

Point 7: remains some doubts about the efficiency of the XTR technology.

Point 9: Ok.

Author Response

Thank you so much for providing your constructive feedback. We have included a list of our responses to the remaining points outlined below. Thank you again.

The authors replied to all the reviewer comments but they did not add some required experiments. In my opinion, they do not increase the scientific soundness of the manuscript.

Point 1: Ok.

We highly appreciate your feedback. Thank you.

Point 2: Ok.

We highly appreciate your feedback. Thank you.

Point 3: Ok.

We highly appreciate your feedback. Thank you.

Point 4: the Authors just added better description of the method to define the degree of crosslinking. However, I would ask a better explanation about the experimental procedure to determine the different degree of crosslinking. 

Thank you very much for your helpful feedback. For this, we have added a more detailed step-by-step process that has been used in the laboratory, as detailed below: “In this study, the determination of the degree of cross-linking of the nominated HA-based fillers was performed using the nuclear magnetic resonance (NMR)-based approach described by Wende and colleagues.22 HA is hydrolyzed using an acidic medium and then, the degradation products are analyzed by one-dimensional NMR. First, the samples were diluted to 4 mg/mL with acid water (HCl 0.1 M, pH 1.5), heated and gently mixed at 75°C overnight. Once cooled, these samples were buffered at pH 7.0 with NaOH 1 M and 0.1 M, were frozen and then lyophilized. After, the samples were dissolved in 0.5 mL of deuterated water (D2O) and then subjected to NMR Bruker 400 mHz analysis for the determination of MoD%, CrR and CrD. With the 1H-NMR set up, 64 scans and a recycle delay (D1) of 10 seconds were used. For the 13C-NMR set up, 8192 scans and a D1 of 10 seconds were used.

The three main parameters were then measured using the following formulae: (1) MoD (%) = (I δH1.5/4)/ (I δH1.9/3) x 100, (2) CrR = 1-IδC62.7/ (Iδ25.2/2), and (3) CrD (%)= (CrR x MoD) x100.”

Point 5: Ok.

We highly appreciate your feedback. Thank you.

Point 6: I believe that the degradation studies are fundamental to define the properties of a HA a filler. Therefore, I continue to say that degradation test is fundamental to increase the scientific soundness of this manuscript.

We appreciate the constructive feedback. The laboratory has conducted an evaluation of the stability of fillers manufactured with XTR technology and we have included some insights in the current version of the manuscript (Section 2.5, 3.4 and 4.4).  

Point 7: remains some doubts about the efficiency of the XTR technology.

Thank you and the feedback is well taken. In this study, we presented as much objective information as possible to aid in the decision of the doctors, given that more and more information on new formulations and cross-linking technologies has been published and thus, may be overwhelming. Our hope is that through this study, users of soft tissue fillers can distinguish which products are appropriate for specific indications for their individual patients. XTR, much like the other fillers featured here, may have its own advantages and disadvantages for certain patient types depending on the filler properties. 

Point 9: Ok.

We highly appreciate your feedback. Thank you.

Round 3

Reviewer 3 Report

The authors replied to all the reviewer comments raising the scientific value of the work and claryfing many aspect of this work.